# Optimization of Fermentation Conditions and Metabolite Profiling of Grape Juice Fermented with Lactic Acid Bacteria for Improved Flavor and Bioactivity

**DOI:** 10.3390/foods12122407

**Published:** 2023-06-18

**Authors:** Dilinu Kuerban, Jing Lu, Zekun Huangfu, Liang Wang, Yanan Qin, Minwei Zhang

**Affiliations:** Xinjiang Key Laboratory of Biological Resources and Genetic Engineering, College of Life Science & Technology, Xinjiang University, Urumqi 830046, China; dln13699358685@163.com (D.K.); jinglu@xju.edu.cn (J.L.); 15253021751@163.com (Z.H.); bluetop23@sina.com (L.W.); zhang78089680@sina.com (M.Z.)

**Keywords:** multi-strain fermentation, metabolomics, UHPLC-QE-MS/MS

## Abstract

To enrich the flavor compounds and retain the content of polyphenolics in grape juice (GJ) under long-term storage, *Lactiplantibacillus plantarum*, *Lactobacillus acidophilus*, *Lacticaseibacillus casei*, and *Lacticaseibacillus paracasei*, were screened and the optimal fermentation conditions were determined as fermentation temperature of 41.2 °C for 24 h with an initial LAB density of 8.5 × 10^6^ CFU/mL. Surprisingly, the retention rates of TPC still remained at 50% after storage for 45 days at 4 °C. Moreover, 251 different metabolites were identified, include 23 polyphenolics, 11 saccharides, and 9 organic acids. Most importantly, the total content of polyphenolics reserved was 92.65% at the end of fermentation. Among them, ephedrannin A content significantly decreased; however, 2′,6′-Di-O-acetylononin gradually increased with the fermentation time, which resulted in FGJ maintaining excellent bioactivity. Meanwhile, organic acid content (palmitoylethanolamide, tetraacetylethylenediamine) increased with saccharides (linamarin) decreasing, which leads to FGJ having a unique taste. Furthermore, a total of 85 Volatile organic compounds (VOCs) were identified, mainly including esters, aldehydes, and alcohols. Interestingly, key VOCs could be formed by carboxylic acids and derivatives, and fatty acyls via complex metabolic pathways.

## 1. Introduction

Fermented juice with lactic acid bacteria (LAB), with high antioxidant activities, which improves gastrointestinal health, and which has antibacterial activities, has attracted more attention, since LAB fermentation could increase the content of phytochemicals and enrich the flavor of juice via catabolism of disaccharides and hydrolysis of phenolic compounds [1,2,3,4,5,6]. Among them, *Lactiplantibacillus plantarum* could significantly increase the flavonol content and antioxidant activity of juice [7], which could be attributed to the reductive conversion of phenols [8] and enzymatic hydrolysis of polymerized phenolic compounds during fermentation [9]. Additionally, *Lactobacillus acidophilus* and *Lacticaseibacillus casei* displayed high a-galactosidase activity, and released isoflavone aglycone during the fermentation process [10]. More interestingly, the fermentation with LAB (i.e., *L. acidophilus*, *L. casei*, *L. helveticus*, and *L. plantarum*) had an obvious influence on the aldehydes, and other volatile organic compounds (VOCs) intensified the fruit aroma via protein and carbohydrate metabolic pathways in the fermentation period [11]. *L. plantarum* could markedly increase the acetic acid in fermented juices through the acetate kinase route of the phosphogluconate pathway [12,13].

Although there are advantages of single-strain fermentation, to synchronize and enrich the bioactive compounds and flavor of fermented fruit juice, multi-strain fermentation is a crucial alternative. Pear juice fermented with three *Lactobacillus strains* (*L. plantarum* 90, *Lactobacillus helveticus* 76, and *L. casei* 37), in single-strain and in mixtures, showed that the colony counts and concentration of volatile compounds of multi-strain fermentation were higher than those of single-strain fermentation [4]. The mixing of three types of *L. plantarum* during the fermentation of bergamot juice displayed better antioxidant activity than single-strain fermentation [14]. Compared with single-strain fermentation, the multi-strain fermentation with microbial complexity enhanced the various metabolites (i.e., polyphenolics, saccharides, organic acids). Therefore, based on the low cost and high efficiency, multi-strain fermentation can be considered an essential option to improve the bioactive compounds and flavor profile of the juice, and to improve the value added and economic value of the fruits.

During the LAB fermentation process, plant metabolites, such as phenolic acids, fatty acids, amino acids, and saccharides, could be converted to VOCs to strengthen the flavor of fermented fruit juice [15]. For example, valine and isoleucine can be converted into α-keto acids during fermentation and then decarboxylated in the corresponding aldehydes, alcohols, and carboxylic acids [16]. Furthermore, key VOCs exhibited significant correlations with amino acids and fatty acyls after being co-fermented by *L. plantarum*, *Lactobacillus rhamnosus* GG, and *Streptococcus thermophilus* [17]. These findings suggest that multi-lactic acid bacteria fermentation significantly affects amino acid and lipid metabolism, leading to VOC formation.

In order to provide insight into the properties of metabolites and VOCs of multi-lactic acid bacteria fermented grape juice (FGJ), the present study produced a multi-strain (*L. plantarum*, *L. acidophilus*, *L. casei*, and *L. paracasei*) FGJ under the optimal fermentation process. Subsequently, the characteristics of various metabolites in GJ during the multi-strain fermentation periods were studied, and we found that the various metabolites played a significant role in the antioxidant characteristics and flavors of FGJ. Finally, VOC profiles were analyzed at the end of the fermentation, and then based on relative odor activity value (ROAV) and network correlation analysis, the key VOCs were determined to be esters and aldehyde, which were associated with fruity and sweet odors and could transfer by carboxylic acids and derivatives, and fatty acyls via complex metabolic pathways.

## 2. Materials and Methods

### 2.1. Materials

Muscat Hamburg grapes were purchased from a local market (Urumqi, Xinjiang, China). Pectinase enzyme (BR, 500 U/mg), cellulase enzyme (BR, 500 U/mg), gallic acid (HPLC ≥ 98% B20106-500 mg), and Folin–Ciocalteu phenol reagent were obtained from Shanghai YuanYe Biotechnology (Shanghai, China). Na_2_CO_3_ (AR 500 g, 99.8%) was obtained from Jiangsu GuoYao Chemical Reagents Company (Beijing, China). Four commercial LAB strains (*Lactiplantibacillus plantarum*, *Lactobacillus acidophilus*, *Lacticaseibacillus casei*, and *Lacticaseibacillus paracasei*) were obtained from Shandong Zhongke Jiayi Biotechnology (Shandong, China). The methanol and acetonitrile were obtained from CNW Technologies (New York, NY, USA). The ammonium acetate was obtained from IGMA-ALDRICH (St. Louis, MI, USA). Ethanoic acid was obtained from Fisher Chemical (Hampton, NH, USA). The 2-octanol (≥99.5%) was obtained from Tokyo Chemical Industry (Tokyo, Japan).

### 2.2. Grape Juice Preparation

Fresh grapes with peels and without seeds were crushed using a crusher (MJ-PB80W2-012). Then, in order to degrade pectin, cellulose, the pectinase enzyme, and cellulase enzyme (pectinase enzyme: 0.1 g, cellulase enzyme: 0.035 g) were added into GJ (500 mL) and then kept at 50 °C for 6 h. After that, GJ was sterilized at 95 °C for 15 min.

### 2.3. Single-Strain Fermentation

After cooling to room temperature, 1 × 10^7^ CFU/mL of the single strains (*L. plantarum*, *L. acidophilus*, *L. casei*, and *L. paracasei*) were inoculated into the pasteurized GJ and fermented at 37 °C for varying times. Then, 50 mL of FGJ was collected in intervals of 8 h (8, 16, 24, and 32 h) for physicochemical analyses of single strain.

### 2.4. Physicochemical Analyses

#### 2.4.1. Determination of Total Phenolics Content (TPC)

The total phenolics content (TPC) was determined based on Folin–Ciocalteu method [18] with slight modifications. Briefly, 5.0 mL water, 1.0 mL coloration, and 3.0 mL 7.5% Na_2_CO_3_ solution were added, respectively, into 1.0 mL extract of samples. Then, optical density at 765 nm was recorded after coloration for 2 h. Results were expressed as gallic acid equivalents (GAE) of samples through gallic acid standard curve calibration (0~0.02 mg/mL).

#### 2.4.2. Determination of Total Soluble Solids (TSS), pH, Clarity

The total soluble solids content (TSS) was determined with Abbe refractor (2WAJ) at 25 °C. The pH of FGJ was determined using a pH meter (PHS-3C). The clarity was measured using a spectrophotometer at 660 nm.

### 2.5. Multi-Strain Fermentation

#### 2.5.1. Optimization of the Proportion of Multi-Strain

After identifying the characteristics of single-strain, to synchronize and enrich the bioactive compounds and flavor of FGJ, the proportion of the multi-strains was determined with a uniform design experiment performed using a U8*(8^4^) uniform design table (Appendix A); *L. acidophilus*, *L. plantarum*, *L. casei*, and *L. paracasei* were set at 8 levels as each factor and the uniform design experiment table was determined with the TPC as the response value.

#### 2.5.2. Optimization of Multi-Strain Fermentation Conditions

To obtain the optimal fermentation conditions, the central composite design using Design-Expert 10.0 software (Stat-Ease, Inc., Minneapolis, MN, USA) was employed to investigate the effects of three independent variables (fermentation time (A), fermentation temperature (B), LAB density (C)) on TPC (Y), and the independent variables were coded at three levels (−1.68, −1, 0, 1, 1.68) (Appendix A).

### 2.6. Metabolite Extraction and UHPLC-QE-MS/MS Analysis of Multi-Strain Fermentation

Based on the optimal proportion, multi-strain were inoculated into pasteurized GJ and fermented at the optimal processing conditions. After, 50 mL FGJ was collected at intervals of 8 h for UHPLC-QE-MS/MS analyses based on a multi-strain growth curve. Eventually, the final sample was collected for HS-SPME-GC-MS analyses.

Metabolite analysis was performed using a UHPLC system (Vanquish, Thermo Fisher Scientific, Waltham, MA, USA) with UPLC BEH Amide column (2.1 mm × 100 mm, 1.8 μm) coupled to an Orbitrap Exploris 120 mass spectrometer (Orbitrap MS, Thermo, Waltham, MA, USA). The mobile phase consisted of 5 mmol/L ammonium acetate and 5 mmol/L acetic acid in aqueous solution (A) and acetonitrile solution (B), self-sampling temperature was 4 °C, and injection volume was 2 μL. The sputtering voltage of the ions was set to 3.8 kV and −3.4 kV for the positive and negative modes, respectively. The sheath gas flow rate and the auxiliary gas flow rate were set to 50 arb and 15 arb, respectively.

### 2.7. VOC Analysis of Multi-Strain Fermentation

#### 2.7.1. Determination of Volatile Compounds

The VOCs were characterized with HS-SPME/GC-MS. Briefly, 500 μL of samples was added into 20 mL headspace bottle, and 10 μL 2-octanol was added as an internal standard. They were incubated at 60 °C for 49 min in a PAL rail system SPME cycle. Subsequently, an Agilent 7890 gas chromatograph system coupled with a 5977B mass spectrometer was used for GC-MS analysis, and the mass spectrogram data were obtained in a scanning mode with a range of 20 to 400 m and a solvent delay of 0 min.

#### 2.7.2. Identification of Key VOCs

The ROAV values were calculated based on the relative concentrations of VOCs at 24 h fermentation of GJ, as shown below:$${\mathrm{ROAV}}_{i}=100\;\times \;\frac{C_{i}}{C_{\max}}\times \frac{T_{\max}}{T_{i}}$$

C_max_ is the compound with the maximum odor activity, and T_max_ is the threshold for C_max_. C_i_ is the concentration of the targeted volatile compound, and T_i_ is the C_i_ odor threshold [19]. Compounds of which the ROAV > 1 are regarded as the key aroma active compound [20].

### 2.8. Statistical Analysis

Differences between groups were assessed using Duncan’s test for multiple comparisons. *p* < 0.05 was taken to indicate a difference that was statistically significant. The multivariate analysis included principal components analysis (PCA) and orthogonal partial least squares discriminant analysis (OPLS-DA), and metabolite values from the experimental groups were filtered on the basis of *p* < 0.05 and variable projection importance (VIP) values. A hierarchical cluster heatmap was generated using the TB tools version 1.082 (Guangdong, China). Pearson correlation (r > 0.5, *p* < 0.05) analysis was carried out using R (version 3.6.3) based on the tools of OmicStudio (https://www.omicstudio.cn/tool (accessed on accessed on 15 November 2022).

## 3. Results and Discussion

### 3.1. Determination of the Strains’ Function and Optimizing the Fermentation Conditions

Considering the cost efficiency, four commercial LAB strains were selected to enrich the flavor compounds and retain the content of polyphenolics in GJ under long-term storage. *L. Plantarum* and *L. acidophilus* exhibited good growth capacities and increased the phenolic content during the fermentation process; additionally, *L. casei* and *L. paracasei* were highly capable of growing and accelerating the fermentation rate [21,22,23]. Hence, the above four strains were inoculated into the FGJ separately, based on the fermentation process detailed in Appendix A. Meanwhile, the important indicators of fermented fruit included the TPC, TSS, LAB density, clarity, and pH of FGJ, which were determined during the various periods (8 h, 16 h, 24 h, 32 h). As expected, after 16 h of fermentation with *L. acidophilus*, the TPC increased from 0.329 to 0.367 GAE mg/mL (increased by 12%) in FGJ (Appendix A), which could be attributed to the *L. acidophilus*-produced glycosidase or esterase leading to the cleavage and acidification of glycosides and esters joined with phenolic acids, making the bound phenols transition to free phenols [24,25]. Moreover, the largest value of TSS was found with *L. acidophilus* at 16 h of fermentation (18.4 °Brix) (Appendix A). However, the clarity, LAB density, and pH showed slight changes when fermented with *L. acidophilus* during the fermentation of FGJ (Appendix A).

Compared with *L. acidophilus*, the *L. plantarum* played a significant role in clarity and LAB density (Appendix A). At a fermentation time of 16 h, the density of LAB reached a maximum of 3.08 × 10^9^ CFU/mL, and the clarity was 85.5%. In addition, fermentation with *L. casei* also showed an impact on the LAB density (3 × 10^9^ CFU/mL) after 16 h of fermentation, due to the *L. casei* having a great ability to grow and steer the fermentation process [15]. With *L. paracasei* fermentation, a slight decrease in pH from 3.80 to 3.70 was observed in the FGJ (Appendix A), which could be related to organic acids (i.e., trifluoromethanesulfonic acid, palmitoylethan-olamide, tetraacetylethylenediamine) produced during the fermentation [16,26], which was also confirmed by UHPLC-QE-MS/MS analysis. These results suggested that GJ fermented with *L. plantarum*, *L. acidophilus*, *L. casei*, *and L. paracasei* may be a kind of product with high nutritional value and good flavor. Therefore, the inoculation proportion of four strains and the fermentation conditions in FGJ were further investigated.

Multi-strain fermentation is a key option to enrich the bioactive compounds and flavors of FGJ. Therefore, the proportion of multi-strain was confirmed by uniform design (Appendix A). The maximum TPC (0.379 mg GAE/mL) was obtained using the strain combinations of *L. plantarum*, *L. acidophilus*, *L. casei*, *L. paracasei* = 28.1%, 28.1%, 23.97%, 19.83%. Finally, the fermentation conditions (temperature, time, and initial LAB density) on TPC were optimized using the response surface methodology. The results indicated that the optimal processing conditions were fermentation temperature 41.2 °C for 24 h with an initial LAB density of 8.5 × 10^6^ CFU/mL (Appendix A), and the TPC reached 0.399 mg GAE/mL. Surprisingly, the retention rates of TPC were still higher than 50% after accelerated storage for 45 days at 4 °C. (Appendix A).

### 3.2. Profiles of Metabolites during the Multi-Strain Fermentation

To gain a better understanding about the profiles of FGJ, the metabolites of GJ fermented in the optimal processing conditions were analyzed. According to the microbial growth curve, the 0 h, 8 h, 16 h, and 24 h, corresponding to the lag stage, logarithmic growth stage, and stationary stage, respectively, were selected to identify metabolites in FGJ using an UHPLC-QE-MS/MS analytical technique during the multi-strain fermentation stages (Appendix A). In our study, a total of 23,879 peaks were extracted, of which 14,815 and 9064 peaks were extracted in the positive and negative modes. In order to improve the validity of data, multivariate statistics techniques (PCA and OPLS-DA models) were employed for a list of processed metabolites for further analysis. As for metabolite diversity, we used clustering by fermentation times distinguishable on the PCA and *p*-values less than 0.002 and 0.001 in ESI^+^ and ESI^−^ modes, respectively (Figure 1 and Appendix A). As shown, there are large diversifications between the unfermented GJ (0 h) and FGJ (8 h, 16 h, 24 h), which elucidated the fact that the LAB strains exhibited diverse growth and metabolic patterns in the fermentation process, resulting in different contents of metabolites and VOCs [27]. However, the variability of FGJ during the 8 h, 16 h, and 24 h was not significant, suggesting that the metabolites in FGJ underwent fewer changes during fermentation for 8 h, 16 h, and 24 h.

The model distinguished the three different fermentation stages (Appendix A), including the *primary fermentation stage* (0–8 h), the *post fermentation stage* (8–16 h), and the *final fermentation stage* (16–24 h). In total, 251 differential metabolites in the three different fermentation stages in the positive and negative ion modes were screened according to VIP > 1.0 and *p* < 0.05. They were mainly classified into amino acids (23), polyphenolics (23), saccharides (11), organic acids (9), steroids (13), fatty acids (6), fatty acyls (6), glycerophospholipids (12), organic heterocyclic molecules (13), terpenoids (6), and alkaloids (8) (Figure 2).

As shown in Figure 2 and Appendix A, 101 differential metabolites were identified in the *primary fermentation stage*, mainly including amino acids (12), polyphenolics (16), saccharides (7), organic acids (3), steroids (7), fatty acids (4), fatty acyls (4), glycerophospholipids (12), organic heterocyclic molecules (9), and terpenoids (4). During the *post fermentation stage*, 52 differential metabolites were identified, of which there were amino acids (8), polyphenolics (5), saccharides (1), organic acids (3), steroids (5), fatty acids (2), fatty acyls (1), glycerophospholipids (12), organic heterocyclic molecules (2), terpenoids (2), and others (17). Meanwhile, 36 differential metabolites were identified during the *final fermentation stage*, including amino acids (3), polyphenolics (2), saccharides (3), organic acids (2), steroids (1), fatty acyls (1), organic heterocyclic molecules (2), and terpenoids (1). Results indicated that different metabolisms existed between the three stages and could be reliably used to screen for differences in metabolites, and the changes in metabolites were greater in the primary fermentation stage than the post and final fermentation stages. In fact, the change in metabolites was related to the role of the strains in different fermentation periods.

To better understand changes in major differential metabolites in FGJ, the key differential metabolites in three different fermentation stages were screened according to *p*-values (less than 0.05), VIP values (greater than 1.0), and FOLD-CHANGE values (greater than 1.5 or less than 0.5) (Appendix A). Specifically, these were polyphenolics, saccharides, and organic acids, which are important contributors to the antioxidant properties and flavor of FGJ.

The total content of polyphenolics reserved was 92.65% at the end of the fermentation process, among which, the ephedrannin A content’s significant decrease was observed (from 0.232 to 0.097 mg/mL) due to LAB strain (i.e., *L. plantarum* 90, *L. casei* 37, *L. paracasei* 01, and others), depolymerized high molecular weight phenolics, or the transformation of individual phenolic compounds [28] (Figure 3A–C). In contrast, the 2′,6′-Di-O-acetylononin content gradually increased with the fermentation time, reaching maximum values at 24 h of fermentation (0.060 mg/mL). The increase in phenolic compounds could be attributed to the capability of *L. plantarum* and *L. casei* to convert simple phenolics and depolymerize the high molecular weight phenolic compounds via hydrolytic enzymes during fermentation [21,29,30]. During multi-strain fermentation, the total content of saccharides increased from 101.893 to 107.224 mg/mL. As can be seen in Figure 3D–F, the 1-Kestose and galactotriose content increased remarkably at the beginning of fermentation, associated with the hydrolysis of sucrose, pectin, and other complex polysaccharides present in GJ [31]. In contrast, the linamarin content declined from 0.138 to 0.106 mg/mL (declined by about 23%) after 24 h of multi-strain fermentation. The reduction in saccharides during fermentation was a result of the *Lactobacillus* using simple carbohydrates as a source of energy for growth [14]. The total content of organic acids was almost constant (from 6.561 to 6.531 mg/mL) during the multi-strain fermentation, however, of which the palmitoylethanolamide and tetraacetylethylenediamine content reached maximum values (0.053 and 0.202 mg/mL, respectively) at end of fermentation (Figure 3H,I). LAB metabolized carbon sources through fermentation, resulting in the formation or degradation of organic acids [32,33]. Meanwhile, trifluoromethanesulfonic acid content decreased; due to this, they could be changed to VOCs (Figure 3G).

As shown in Figure 4, during the *primary fermentation stage*, the 2′,6′-Di-O-acetylononin were positively correlated with 1-Kestose and galactotriose, which could be due to the *L. plantarum* removing the saccharide moieties and hydrolyzed galloyl moieties of a variety of phenolic compounds during fermentation, leading to phenolics being released [34]. Additionally, at the *final fermentation stage*, the tetraacetylethylenediamine was negatively correlated with linamarin (Appendix A); because of this, simple saccharides might be metabolized to converse into organic acids during LAB fermentation [35]. These results show that there is a certain correlation among the metabolites, which may improve the metabolic activity and affect the functional bioactive compounds and flavor of the FGJ.

### 3.3. Analyses of VOC Profiles

The flavor profiles are one of the most significant characteristics of fruit juice for consumption [33]. The VOCs of FGJ were investigated at the end of the multi-strain fermentation (24 h) through an HS-SPME-GC-MS approach. It is worth noting that a total of 85 VOCs were identified; they were classified into 26 esters, 16 aldehydes, 14 ketones, 13 alcohols, 1 acid, and 15 others (Appendix A). The results deduced that multi-strain fermentation was able to produce plenty of VOCs in FGJ, due to the *L. acidophilus*-, *L. casei*-, and *L. plantarum*-unique metabolic pathways [10].

In order to screen key VOCs, according to Section 2.7.2, the ROAV was determined, and eight key VOCs with ROAV > 1 were identified, including esters (butanoic acid, 2-methyl-, ethyl ester, butanoic acid, ethyl ester, hexanoic acid, ethyl ester, butanoic acid, 3-methyl-, ethyl ester) and aldehydes (nonanal, heptanal, hexanal, butanal, 3-methyl-) (Table 1). These were associated with green, sweet scents that may be responsible for improving FGJ’s aroma profile [17].

### 3.4. Association Analysis between the VOCs and Non-Volatile Compounds

To better elucidate the relationship between VOCs and non-volatile compounds of FGJ, the association of saccharides (the top 12 most abundant which mean relative abundance > 0.5%) (Appendix A), carboxylic acids and derivatives, and fatty acyls (the top 27 most abundant, which means relative abundance > 0.5%) (Appendix A) with key VOCs (ROAV > 1) (Table 1) were analyzed using Pearson correlation (r > 0.5, *p* < 0.05). Interestingly, there was no significant correlation with saccharides (Appendix A); however, they displayed as being significantly associated with carboxylic acids and derivatives, and fatty acyls (Figure 5).

During the whole fermentation period, the nonanal and butanoic acid, 3-methyl-, ethyl ester were correlated with different amino acids and fatty acids. Figure 5A,B shows that the L-Allothreonine positively correlated with butanoic acid, 3-methyl-, and ethyl ester, while being negatively correlated with hexanoic acid and ethyl ester. A positive relationship was represented between nonanal and *Lauroyl diethanolamide* (Figure 5A), because nonanal can be formed via β-oxidation of unsaturated fatty acids [17]. Overall, these findings illustrated that LAB utilized carboxylic acids and derivatives, and fatty acyls to form key VOCs via complex metabolic pathways that provide FGJ with a unique flavor during multi-strain fermentation.

## 4. Conclusions

In this study, FGJ was prepared via multi-strain fermentation by four strains (*L. plantarum*, *L. acidophilus*, *L. casei*, *L. paracasei* = 28.1%, 28.1%, 23.97%, 19.83%) at 41.2 °C for 24 h with an initial LAB density of 8.5 × 10^6^. Further, 251 different metabolites were identified during the three different fermentation periods, of which 9 key differential metabolites (2′,6′-Di-O-acetylononin,5,7-dihydroxy-2-phenyl-6-[3,4,5-trihydroxy-6-(hydroxymethyl)oxan-2-yl]-8-(3,4,5-trihydroxyoxan-2-yl)-4H-chromen-4-one, ephedrannin A, 1-Kestose, galactotriose, linamarin, trifluoromethanesulfonic acid, palmitoylethan-olamide, tetraacetylethylenediamine) that affect the bioactivity and flavors of FGJ were selected. Moreover, 85 VOCs and 8 key VOCs (butanoic acid, 2-methyl-, ethyl ester, butanoic acid, ethyl ester, hexanoic acid, ethyl ester, butanoic acid, 3-methyl-, ethyl ester, nonanal, heptanal, hexanal, butanal, 3-methyl-) were identified after fermentation for 24 h. Additionally, the correlation of 27 carboxylic acids and derivatives, and fatty acyls with eight key VOCs, which could provide pleasant aromas (flower, green, and fruit odors), was illustrated, in which LAB used arboxylic acids and derivatives, and fatty acyls to change or form VOCs during the multi-strain fermentation through complex metabolic pathways. Overall, this study offers new perspectives for the future research of fruit and vegetable fermentation with multi-strain.

## Figures and Tables

**Figure 1 foods-12-02407-f001:**
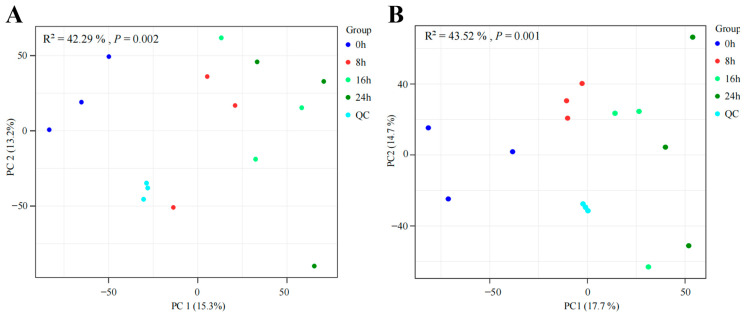
Principal component analysis (PCA) score plots of FGJ during different multi-strain fermentation stages ((**A**) principal component analysis in positive mode; (**B**) principal component analysis in negative mode).

**Figure 2 foods-12-02407-f002:**
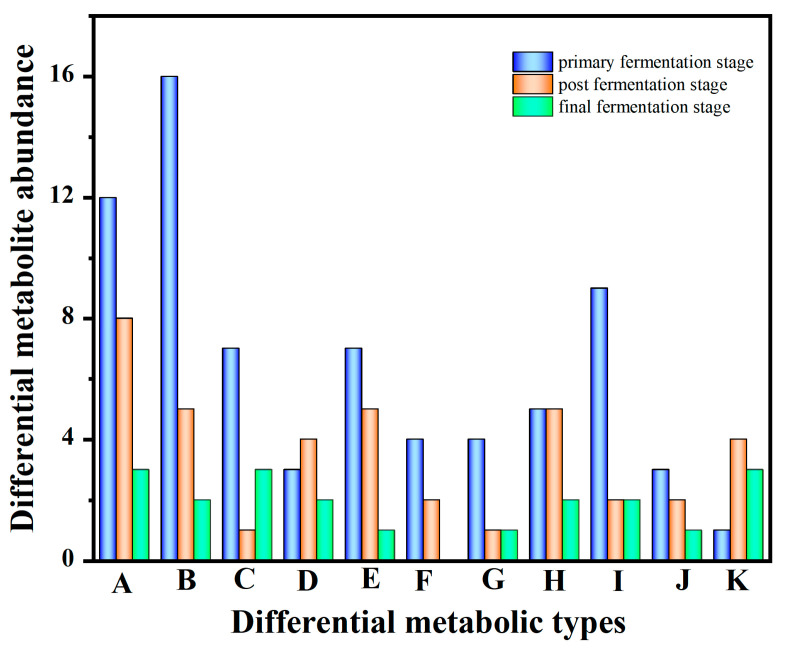
Changes in differential metabolites in FGJ during different multi-strain fermentation stages (A: amino acids; B: polyphenolics; C: saccharides; D: organic acids; E: steroids; F: fatty acids; G: fatty acyls; H: glycerophospholipids; I: organic heterocyclic molecules; J: terpenoids; K: alkaloids).

**Figure 3 foods-12-02407-f003:**
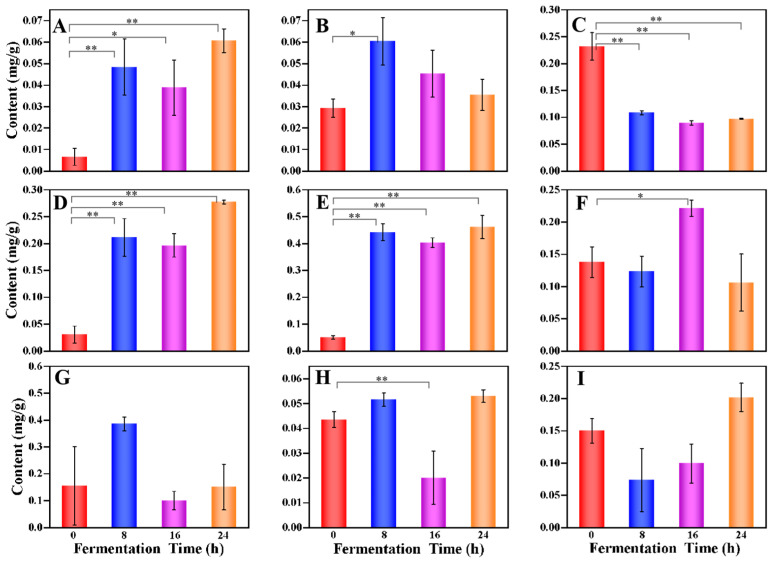
Dynamic changes of nine key differential metabolites in FGJ during different multi-strain fermentation stages ((**A**) 2′,6′-Di-O-acetylononin; (**B**) 5,7-dihydroxy-2-phenyl-6-[3,4,5-trihydroxy-6- (hydroxymethyl)oxan-2-yl]-8-(3,4,5-trihydroxyoxan-2-yl)-4H-chromen-4-one; (**C**) ephedrannin A; (**D**) 1-Kestose; (**E**) galactotriose; (**F**) linamarin; (**G**) trifluoromethanesulfonic acid; (**H**) palmitoylethan- olamide; (**I**) tetraacetylethylenediamine). Notes: * represents a significant difference, *p* < 0.05; ** represents a very significant difference, *p* < 0.01.

**Figure 4 foods-12-02407-f004:**
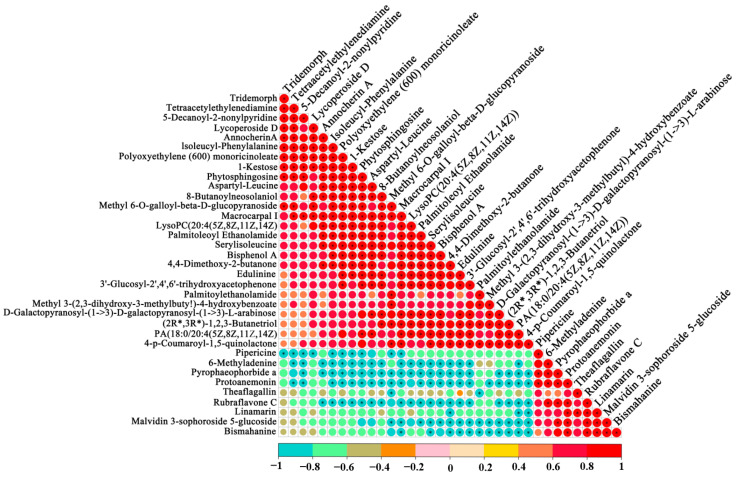
Correlation maps of metabolites in fermented grape juice during final fermentation stage.

**Figure 5 foods-12-02407-f005:**
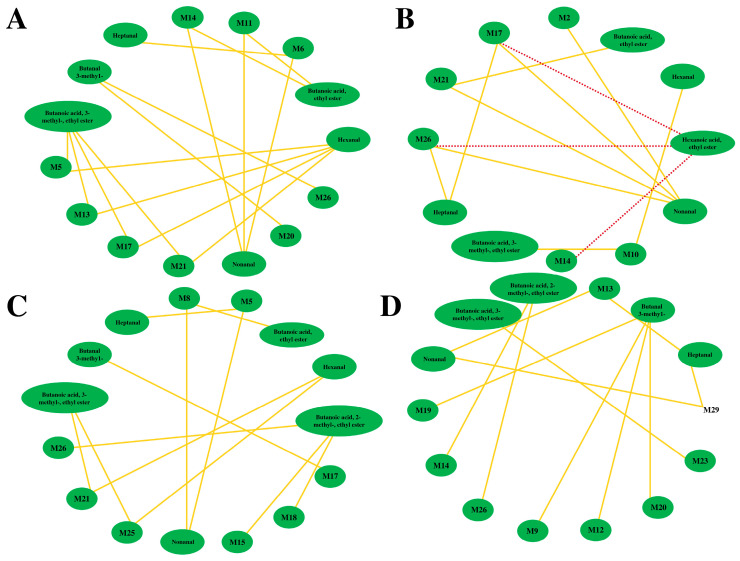
Networks between the VOCs and carboxylic acids and derivatives, and fatty acyls in FGJ based on correlation analyses during the different fermentation times ((**A**) 0 h; (**B**) 8 h; (**C**) 16 h; (**D**) 24 h). Notes: Solid lines represent positive correlation and dashed lines represent negative correlation.

**Table 1 foods-12-02407-t001:** ROAV and odor description of key VOCs in FGJ after 24 h of multi-strain fermentation.

No.	Compounds	^a^ Odor Threshold (mg/L)	^b^ Odor Description	Relative Odor Activity Value (ROAV)
V1	butanoic acid, 2-methyl-, ethyl ester	0.000063	Apple	100.00
V2	butanoic acid, ethyl ester	0.003	Banana, pineapple, strawberry	15.026
V3	hexanoic acid, ethyl ester	0.0022	Banana, fruit, apple peel	6.236
V4	nonanal	0.008	Fat, citrus, green	1.226
V5	heptanal	0.0028	Fat, citrus, rancid	1.463
V6	butanal, 3-methyl-	0.0012	Ethereal, aldehydic	1.505
V7	butanoic acid, 3-methyl-, ethyl ester	0.00011	Fruity, sweet, apple	1.588
V8	hexanal	0.073	Green, tallow, fat	3.394

^a^ Odor thresholds (mg/L) were taken from previous works [19]; ^b^ odor descriptors were cited from www.flavornet.org (accessed on 1 November 2022).

## Data Availability

Data is contained within the article and Appendix A.

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
