# Peer review of "Optimization of Fermentation Conditions and Metabolite Profiling of Grape Juice Fermented with Lactic Acid Bacteria for Improved Flavor and Bioactivity"

_foods, 2023, doi:10.3390/foods12122407_

Round 1

Reviewer 1 Report

The manuscript “Insight into the properties of metabolites and volatile organic compounds of multi-lactic acid bacteria fermented grape juice” provides a novel Optimization of Fermentation Conditions and Metabolite to Grape Juice. This research content has the application value. So, I suggest changing title to: “"Optimization of Fermentation Conditions and Metabolite Profiling of Grape Juice Fermented with Lactic Acid Bacteria for Improved Flavor and Bioactivity"

I just have some comments:

 1. The references at introduction should be updated

2. line 90: 2.3. Determination of total phenolics content (TPC), total soluble solids (TSS), pH, clarity, and  LAB density, need more details  

 3. all methods as a general need more details    

4. where’s attached Supplementary Figure and Tables

Moderate editing of English language required

Reviewer 2 Report

I am very grateful you for the invitation to review manuscript foods-2425419 by Kuerban and coauthors "Insight into the properties of metabolites and volatile organic compounds of multi-lactic acid bacteria fermented grape juice”. This study produced a multi-strain (L. plantarum, L. acidophilus, L. casei, and L. paracasei) FGJ under the optimal fermentation process. Subsequently, the characteristics of various metabolites in GJ during the multi-strain fermentation periods were studied and found that the various metabolites play a significant role in antioxidant characteristics and flavors of FGJ. Finally, VOCs profiles were analyzed at the end of the fermentation, and then based on relative relative odor activity value (ROAV) and network correlation analysis, the key VOCs were determined esters and aldehyde, which were associated with fruity and sweet odor and could transfer by carboxylic acids and derivatives, and fatty acyls via complex metabolic pathways. The work is interesting but needs adjustments to increase the quality of the material.

Comments:

- Abstract: Include a brief sentence about the importance of the study (problem to be solved). The “flavor and bioactives of grape juice” it's a problem?

- Include a more accurate conclusion about the work.

- Line 21: Change the repeated keywords by different words from the title.

- Lines 24-26: Please highlight more clearly the need for this process.

- Line 39: highlight the “potential benefit for human health”.

- Lines 39-40: This sentence should be improved.

- Lines 47-48: Conduct a technical and financial feasibility discussion regarding the benefits of fermentation.

- Introduction: Indicate consumer market for this type of product and total market.

- Line 76: and the “2-octanol ( ≥ 99.5%)”?

- Line 82: Standardize units throughout work (6H -> 6h).

- Sample preparation for determinations is unclear.

- Lines 135-136: Change “figure 1” to “Figure 1”.

- Line 134 and methodology: Indicate whether the process adopted was continuous or a pool of microorganisms.

- Line 140: Which enzymes are listed in the sentence “produced certain enzymes”?

- Line 150: Are the small pH changes related to the already acidic conditions of the product?

- The characteristics of microorganisms, as well as the changes caused by them.

-  Results: The relationship and presentation of technical and economic characteristics must be discussed.

- Lines 183-185: Highlight the metabolisms associated with each stage of fermentation.

- Line 199: Change “table 6” to “Table 6”.

- Lines 201-202: Indicate the main metabolites in each of the stages.

- Line 242: Change “figure 4” to “Figure 4”.

- Line 255: Change “[28]. the VOCs” to “[28]. The VOCs”.

- Lines 291-293: And what is the conversion? Would it be harmful to the product?

- Lines 310-312: This is not so clear. Complementary studies, including sensory studies, are needed.
